# A Different Processing of Time-Domain Induced Polarisation: Application for Investigating the Marine Intrusion in a Coastal Aquifer in the SE Iberian Peninsula

**DOI:** 10.3390/s23020708

**Published:** 2023-01-08

**Authors:** Jesús Díaz-Curiel, Bárbara Biosca, Lucía Arévalo-Lomas, María Jesús Miguel

**Affiliations:** 1Department of Energy and Fuels, School of Mines and Energy, Universidad Politécnica de Madrid, C/Ríos Rosas 21, 28003 Madrid, Spain; 2Ministerio de Ciencia e Innovación España, Paseo de la Castellana 162, 28046 Madrid, Spain

**Keywords:** marine intrusion, induced polarisation, polarizability, decay time

## Abstract

This study presents the developments regarding the time-domain induced polarisation method as a supporting tool for resistivity soundings during investigations of coastal detrital aquifers that are salinized by marine intrusion. The interpretation of resistivity measurements in such aquifers, which have variable hydrochemistry and lithology, involves uncertainties owing to the presence of low-resistivity lithologies, such as clays. To reduce these uncertainties, the use of other geophysical parameters is necessary; hence, this study focuses on induced polarisation since it can be measured simultaneously with resistivity. In detail, we propose the determination of induced polarisation using 1D techniques while developing a different algorithm for processing the induced polarisation data. The aim is to extend the results of this phenomenon, using, instead of chargeability, the concepts of polarisability and decay time, which are extracted from the decay curve, given that they represent more intrinsic properties of the various analyzed subsurface media. Results were obtained by applying this methodology to a Quaternary aquifer of the Costa del Sol in the SE Iberian Peninsula (in the province of Almería) during two different campaigns, one in mid-autumn and one late winter (i.e., in October and February, respectively) are presented. The results reveal the position of the saline front during each campaign while reflecting the seasonal movement of the marine intrusion.

## 1. Introduction

The geological and lithological characteristics of a formation assigned solely based on resistivity can be partially ambiguous when media with similar resistivities coexist. Besides the lithological differences, the resistivity values of geological formations depend on a large number of petrophysical variables in these media as well as the hydrochemical characteristics of the formation water. Many 1D geoelectric prospecting studies often consider that an adequate geological interpretation can only be achieved based on previous knowledge of the area; however, this consideration may not be appropriate as a general criterion.

The intrinsic parameter that best discriminates the media from a geoelectrical point of view is the form factor (*F* = *ρ*/*ρ*_W_, where *ρ* is the resistivity of the medium and *ρ*_W_ is the resistivity of the pore fluid); hence, the hydrochemical characteristics of the pore fluid strongly affect the resistivity of the medium. For example, an aquifer composed of granular lithologies can have resistivities very similar to those of clays (~10 Ω m) when the formation fluid has a conductivity of a few tenths of S/m, corresponding to that of a freshwater/seawater transition fluid (interface). However, this ambiguity can be reduced by using more than one geophysical parameter.

The induced polarisation (IP) survey method allows the acquisition and measurement of a second parameter with the same electrode arrays as those used for obtaining the resistivity. This method is based on the IP phenomenon discovered by Schlumberger in 1920 [1,2] and has been the product of continual development [3,4,5,6]. The method involves analyzing the electrical response of the terrain to the variation of a certain generated electric field, either by measuring over time the field—i.e., the potential—that remains in the medium when the applied current is cancelled (time-domain TDIP) or by measuring the electrical response of the terrain when the frequency of an alternating current varies (frequency domain SIP).

In time-domain measurements, two parameters can be distinguished according to the historical evolution of this method. Initially, the objective was to register the potential immediately after cancelling the current injection, thereby establishing the polarization capacity of the medium through the relationship between the potential measured just before cancelling the applied field *V*_P_ (primary potential) and the potential measured just after cancelling the external field *V*_S_ (secondary potential) ([7]; Equation (1)):(1)$$m=\frac{V_{S}}{V_{P}}\;in\;\left(\frac{mV}{V}\right)$$

However, partly because it is difficult to measure the established *V*_S_, it is common practice to measure the decay of the voltage over time, which is typically measured in successive time windows, while obtaining the value of *m* using the following expression [8]:(2)$$m=\frac{\int_{\;t_{1}}^{\;t_{2}}\;V_{S}\left(t\right)\;\cdot \;dt\;}{\;\left(t_{2}-t_{1}\right)\;\cdot \;V_{P}\;}=\frac{\;\bar{V_{S}}\;}{V_{P}},\;$$
where *V*_S_(*t*) is the potential value measured over time after cancelling the primary potential, and (*t_2_* − *t*_1_) = Δ*t* is the integration interval. Thus, the value ${\bar{V}}_{S}$ obtained through this integral is attributed to the secondary potential.

Although the term *polarisability* was also being used initially, the term *chargeability* prevailed in time and is assigned indistinctly to the parameter obtained by either Equation (1) or (2). Fiadanca et al. [9] differentiated the parameters obtained by Equations (1) and (2) by calling them *intrinsic chargeability* and *integral chargeability*, respectively. The main limitation that the authors identified regarding Equations (1) and (2) was that they might provide decay curves with the same chargeability but different polarisabilities and decay times, suggesting that they correspond to different IP sources (Figure 1). Because IP decay curves do not necessarily correspond to mono-exponential behaviour, the decay time reflected in Figure 1 corresponds to *V*_s_(0)/e. For this reason, in this study, the term *chargeability* refers to the value obtained by Equation (2), and *polarisability P* refers to the value obtained by Equation (1), modified to consider only the part of the secondary potential that corresponds to the *V*_IP_(*t* = 0^+^) polarisation effect.

Some studies have focused on the decomposition of the decay curve at each depth of investigation into a sum of different exponentials. In this study, we present an alternative decomposition involving dividing the decay curve into only three exponentials. One of the three exponentials corresponds to the electromagnetic induction effect; another represents the IP decay due to the different layers influencing each depth of investigation, and a third corresponds to a residual potential found in many of the cases studied.

Regarding the IP in the frequency domain, owing to the increased operability and electronic ease, there has been greater development of these methods. This development began with the investigation of the frequency effect (FE) based on the resistivity variation for two highly differentiated frequencies *ω*_1_ >> *ω_2_* given by Equation (3):(3)$$FE=\frac{\;\left|\;\rho \left(\omega_{2}\right)\;\right|-\left|\;\rho \left(\omega_{1}\right)\;\right|\;}{\left|\;\rho \left(\omega_{1}\right)\;\right|}\;\cdot 100\%.\;$$

This frequency effect has often been calculated by considering *ω*_2_ = 0, i.e., *ρ*(*ω*_2_) = *ρ*_DC_ [10]. In the model defined by Cole and Cole [11], the complex resistivity for frequency *ω*_k_ is given by Equation (4) [12]:(4)$$\rho \left(\omega_{k}\right)=\rho_{0}<1-m\;\{1-{\left[1+{\left(i\;\cdot \;\omega_{k}\cdot \;\tau \right)}^{c}\right]}^{-1}\}>,\;$$
where *ρ*_0_ is the resistivity in direct current, *m* is the chargeability, *τ* is the time constant of the IP, and *c* is the frequency exponent (slope on the logarithmic scale of both sides of the phase curve) with a maximum value of 1. Regarding the relationship between the two domains, which was introduced by Cole and Cole [11], different researchers have introduced different expressions established by different researchers [13,14], including an expression accounting for the electromagnetic coupling (EM) effects [15]. Thus, several studies used the IP method in the frequency domain and its relation to the time domain, thereby determining different time constants depending on the petrochemical and hydrochemical characteristics of the subsurface. Pelton et al. [12] conducted an exhaustive study of the electrical response of metallic formations and found typical frequency-exponent values ranging from 0.1 to 0.6, with a mean value of 0.25; they also conducted an analysis of the spectral behaviour of the studied reservoirs, thereby determining different values for *τ*. Given the above-mentioned terminological differentiation, the logical interrelation of IP in the time and frequency domains is more closely linked to chargeability. 

The aim of this study is to propose a different methodology for the processing of time-domain-induced polarisation. This methodology is applicable to any study of induced polarisation. As an application case, a marine intrusion problem was chosen, specifically in the Almuñécar aquifer, where the intrusion problem has been well-known for years.

After establishing the importance of IP in the time domain, this study introduces in detail a way to process the TDIP decay curves while providing the value of the *V*_IP_(*t* = 0^+^) polarisation potential with the aim of differentiating the presence of clay levels through geoelectric soundings conducted in coastal detrital aquifers. The inclusion of such clay levels, which have higher polarizability values than those of granular strata, improves the inversion of resistivity soundings in which this presence was not considered [16] and elucidated the marine intrusion progress. The proposed methodology was applied to a case of marine intrusion and eventually provided the seasonality of the salt wedge.

One aspect of this study that should be noted is the choice of 1D techniques. Currently, most research teams using electrical prospecting techniques have opted for 2D or 2D + 1 techniques with higher lateral resolution. In this respect, it is worth mentioning the study of Hönig and Tezkan [17], who conducted an exhaustive analysis involving tomographic inversion (2D for 1D inversion and 3D for 2D inversion), while considering *τ* and *c* for cases with *m* > 0.15.

Regarding IP measurement, there are certain drawbacks regarding the tomographic profiling measurement. First, the use of non-polarisable electrodes for conducting potential measurements is not suitable for the current circuit; this is only possible with some of the existing tomography systems or with the use of specific measurement arrays that differentiate the injection positions from the potential-measurement positions. Second, unlike the multi-conductor cables used in tomographic measurement equipment, obtaining higher-quality measurements requires the use of separate cables for current and potential circuits [18]. Third, when investigating coastal aquifers in countries with high proportions of buildings and/or agricultural plots, tomographic measurements are hindered, with the presence of fences and small buildings preventing the use of cables with fixed connectors at certain distances and requiring the use of single-conductor cables with highly variable lengths for different electrode positions.

Marine intrusions that have the greatest repercussions on groundwater exploitation have high spatial extensions; this makes them appropriate for conducting 1D studies. The aim of the project to which these studies belong was to evaluate the temporal evolution of the freshwater/saltwater interface by conducting successive low-cost campaigns, thereby also conditioning the choice of 1D studies with a reduced number of measuring arrays. In relation to 2D studies, the exhaustive work of Ogilvy et al. [19] is worth mentioning since they developed an automated measurement system (ALERT) for monitoring, among other phenomena, the marine intrusion advance and its effects on the underlying Quaternary aquifer.

## 2. Background

The IP prospecting method was initially called over-voltage and became very important in the 1950s for locating metal deposits [20,21,22,23]; its use for exploring clay strata, characterizing petrophysical properties, and investigating groundwaters of different salt concentrations has since then increased.

The initial identification of the influence of clays on the IP responses resulted in several studies on various related theoretical aspects [7,24], which was followed by multiple studies on their application in different environments [25,26,27,28]; others studied the specific characteristics of the responses of clays [29,30,31]. Relationships between the IP parameters and different granulometric characteristics of different media were also established. On the one hand, Börner et al. [32] and Slater and Glaser [33] determined different empirical relationships between specific pore-surface areas and IP results; on the other hand, several studies demonstrated a clear relationship between the decay time, derived from the Cole–Cole model from frequency-domain measurements, and the pore size of the formation. Titov et al. [2], Scott and Barker [34], and Binley et al. [35] presented empirical relationships established between the time constant and minor grain size.

Besides the lithological characteristics, the hydrochemical characteristics can also result in ambiguities in the IP values of different formations in the subsurface. However, the weight of the hydrochemical characteristics is different from those of the lithological and petrophysical characteristics [36], thereby reducing the ambiguities when comparing the resistivity and polarisability of a certain medium. Consequently, many studies have tried to confirm the relationship between IP and the hydrochemical characteristics of formation [37,38,39,40,41]. Such ambiguities occur when studying areas with fully or partially salinized aquifers, which can be confused with clay levels if only resistivity values are available. The ambiguities can be reduced or avoided through IP measurements, which are widely used in studies on marine intrusions and aquifer salinization [42,43,44,45,46,47,48,49]. However, IP measurements have also been applied in recent studies of organic fluid pollution with non-aqueous phases, with the results being rather variable and, in some cases, unsatisfactory [50,51,52,53,54,55,56].

### 2.1. Influence of Electromagnetic (EM) Coupling

As with any measure in a variable electric field, EM induction effects in the subsurface may also be included when conducting IP measurements. Depending on the array used and, in our case, the study time, the relative proportions of the effects may vary. For this reason, the arrays used for IP measurements are usually linear, and the time windows for measuring the IP decay curve are between 0.25 and 2.50 s. Nevertheless, it is necessary to eliminate the possible influence of EM effects on the IP measurements.

The separation of IP effects from EM coupling has been performed more extensively in the frequency domain, where much shorter decay times are considered based on the transfer function.

Luo and Zhang [57] used an equivalent ‘second order’ model to describe the presence of EM coupling in the IP by adding another term to Equation (4):(5)$$\begin{array}{c} \rho \;\left(i\;\cdot \;\omega_{k}\right)=\rho_{0}\;<\;1-m\;\{\;1-{\left[\;1+{\left(i\;\cdot \;\omega_{k}\;\cdot \;\tau \;\right)}^{c}\right]}^{-1}\}-m\prime \;\{\;1-\\ {\left[\;1+{\left(i\;\cdot \;\omega_{k}\;\cdot \;\tau \prime \;\right)}^{c\prime }\right]}^{-1}\}>.\; \end{array}$$

The main theoretical drawback of this technique lies in the variation of resistivity with frequency, while its application in the field is limited by induction phenomena and residual potential, resulting in nonzero signal background.

### 2.2. Measurements in the Time Domain

By studying the decay curve in the time domain, the different effects appearing in it (e.g., induction, polarisation, coupling, noise, etc.) can be more easily discriminated and separated.

The most detailed mathematical process for decomposing the decay curve into a sum of exponentials is the Laplace decomposition. Although this decomposition can result in a very large number of exponentials corresponding to differential decay-time increments, for IP decay curves, the number of exponentials should be limited to the number of studies (time) windows used during the measurement. One of the main advantages of the Laplace decomposition is the identification of areas within the integration period with a higher occurrence of errors or noise, which in the frequency domain would appear as part of the total spectrum. During analysis of the complete curve, the sum of exponential functions fitted to the points may not sufficiently represent some of them, especially if measurements have been performed with a low *V*_p_ value; for these ‘noisy’ bands, specific criteria can be adopted. Overall, it is worth remembering that using filters for non-infinite signals involves limitations.

Other studies have conducted the same type of decomposition in different manners. Luo and Zhang [57] proposed the decomposition of the decay curve *E*(*t*) into a wide series of decreasing exponentials *A*_n_·exp(−*α*_n_·*t*), corresponding to the different contributions to the total polarisation of the different subsurface formations:(6)$$E\left(t\right)=E_{0}\;\left[\;1-\underset{n}{\sum }A_{n}\;\cdot \exp\left(-\alpha_{n}t\right)\;\right],\;\;$$
where $E_{0}=\underset{t\to 0}{Lim}E\left(t\right)$.

Some authors have considered specific coefficients and time constants as functions of different lithologies and different proportions of dissolved ions present in the formation of water. Thus, Xiang et al. [58] considered for the case of 20% pyrite and 80% andesite in 5% 0.01 N ClNa water the following coefficients (Equation (7)):(7)$$\begin{array}{c} E\left(t\right)=E_{0}\;[\;1-0.29\exp\left(-5.7\cdot t\right)-0.11\exp\left(-40\cdot t\right)\\ -0.10\exp\left(-300\cdot t\right)].\; \end{array}$$

However, the coefficients used to separate the EM coupling effects from the IP are defined in the direct solving, but not inversely, for the different decay curves.

Tong et al. [59] conducted laboratory experiments for which the precision of the measurements allowed the relaxation-time spectrum to be considered as the weight coefficients of each of the exponentials into which the decay curve can be decomposed (Equation (8)):(8)$$y\left(t\right)=\frac{V\left(t\right)}{V\left(0\right)}\;\int_{tmin}^{tmax}V_{N}\left(\tau \right)\exp\left(\frac{t}{\tau }\right)d\tau .$$

Apart from a complex acquisition of measurements over a large number of time windows during field surveys, this division requires the almost total absence of noise in the measurements. If the measurements are not sufficiently accurate, the Laplace decomposition may produce a sum of exponentials that do not represent subsurface stratification.

Another method for decomposing the polarisation curve into a sum of two decaying exponentials involves applying the stretched exponential relaxation function [60], although the choice between the two options depends on the physics of the process itself. Some authors have applied this methodology for the IP analysis as an alternative to the Cole–Cole model [61,62].

In this study, we present an alternative decomposition involving dividing the decay curve into only three exponentials: one exponential corresponding to the EM effect, another representing the IP decay due to the different layers influencing each depth of investigation, and a third corresponding to a residual potential that is evident in each case. In our opinion, the latter should also be previously removed in the Laplace decomposition if stratigraphically meaningful results are to be obtained.

## 3. Materials and Methods

One of the main challenges in the application of IP in the time domain is to obtain as reliable data as possible in each time window. Most equipment does not have the possibility of separate stacking of responses for each time window. This is particularly relevant if it is considered that the use of each window implies the inclusion of noise of different frequencies. The equipment used in this study did have this possibility, as well as allowing a higher power-supply intensity than that supported by another current DC equipment.

The methodology for processing decay curves is another key issue in the application of IP in the time domain. The main methodological contribution of this study is the proposal of such processing involving dividing the IP curve into three ‘potentials,’ one potential with a short time period corresponding to the EM effects, another with an intermediate time period corresponding mainly to the IP effects, and a third residual one with much higher time constant (→∞) that is different from those theoretically considered. The reason for including a residual potential is the presence of an electrode potential which, although time-varying, can be considered constant in conventional measurement periods in IP studies. Although this effect has been minimized using non-polarisable electrodes (Figure 2), its elimination is indispensable and relatively easy in the treatment of time-domain IP.

### 3.1. Data Acquisition

The instrumentation consisted of a Scintrex TSQ-3 (Scintrex, Vaughan, ON, Canada) (3 kW) square-wave transmitter with 4 s charging pulses connected to two steel electrodes. With the Scintrex IPR-10A receiver, measurements were obtained in six 520-ms time windows between 260 and 3380 ms (average times from 520 up to 3120). For the ΔV measurement, fat non-polarisable electrodes from Geotron were used, which consisted of semi-permeable porous vessels filled with a supersaturated copper sulphate solution (Figure 2). These electrodes were used to minimize the electrode polarisation effects, which are typically present in IP measurements. Resistivity and IP measurements were conducted simultaneously. The use of non-polarisable electrodes and separate current and potential cables, as mentioned in the introduction, allowed measurements of satisfactory quality. For the same reason, we made sure to place the potential cables perpendicular to the current cables.

### 3.2. Analysis of the Decay Curves

As mentioned before, the processing of the decay curves involves dividing them into three ‘potentials’, one potential with a short time period corresponding to the EM effects, another with an intermediate time period corresponding mainly to the IP effects, and a third residual one with much higher time constant (→∞) that is different from those theoretically considered. In practice, with a fit to the sum of the first two exponential functions, plus a residual value towards which the decay curve is asymptotic, the phenomenon can be considered sufficiently described, with the precision obtainable in the field (Equation (9)):(9)$$V_{s\left(t\right)}=V_{0\mathrm{EM}}{\;\cdot \;e}^{-t/\tau_{\mathrm{EM}}}+V_{0\mathrm{IP}}{\;\cdot \;e}^{-t/\tau_{\mathrm{IP}}}+V_{R},\;$$
where the first term is mainly due to the inductive response of the medium, and the second term is due to the depolarisation of the medium. Note that the time windows used in IP do not adequately estimate the initial induction potential; therefore, the *V*_0*EM*_ value is only analyzed with regard to the decomposition of the decay curve. The reason for including a residual potential *V_R_* is the presence of a potential difference between the electrodes, which can be considered constant in conventional measurement times in IP studies. In fact, *V_R_* showed differences between successive measurement points or stations of each sounding without stratigraphic justification. Although this effect has been minimized by using non-polarisable electrodes (Figure 2), its elimination is necessary and relatively easy in the treatment of IP in the time domain.

Moreover, even when the removal of EM electromagnetic and residual effects are highly reliable, the remaining curve still includes the effects of two factors working in tandem: the salinity of the formation water and the petrological characteristics of each stratum. For this reason, two different parameters were obtained from the IP curve: polarisability P (the extrapolated value of the polarization potential at *t* = 0^+^ divided by the potential just before that time, *t* = 0^−^) and the period or decay constant *τ*_IP_ of the curve. In this study, we considered that the petrology of the medium mainly affects the decay period, whereas the mineralogical characteristics particularly affect the polarisability. As mentioned above, the measurements of polarisability and decay time represent a clear advantage over the measurement of chargeability within a given time window (although this is currently the most common procedure). It is easy to see that two different decay curves can have the same chargeability, measured in a single window; however, analysis of the complete decay curve shows that both the cut-off point at *t* = 0^+^ and the decay period are different while representing more intrinsic values of the rock (Figure 1).

The key of the decomposition, as shown in Equation (9), is twofold: On the one hand, the decomposition of a gradually decreasing signal into a sum of an exponential and a constant value does not represent any remarkable complexity; on the other hand, the decay times *τ_IP_* and *τ_EM_* are intrinsically different. Typically, the polarization time varies between one and several seconds, whereas the induction time is a few tenths of a second long. In this study, the decomposition method involved performing a series of approximating iterations, which required an initial decomposition to avoid convergence problems. Thus, the steps for the decomposition of the decay curve were as follows.

Initially, by minimizing the approximation error of the measured decay-data curve (*V*_n_,*t*_n_) to a simple exponential plus a constant potential, *V*_S(*t*)_ = *V*_0_·e^−(*t*/τ)^ + *V*_R0_, we obtained the value of the cut-off potential *V*_0T_ and verified that the obtained valued coincided (i.e., ±2%) with that obtained by determining the residual potential minimizing the variance between successive point-to-point decays. Then the next steps were:

(1) The approximation of the decay-data curve of a modulated exponential is given by Equation (10):(10)$$V_{S\left(t\right)}=V_{0M}\;\cdot \;e^{-{\left(t/\tau_{M}\right)}^{\;m}}\;,\;$$
through an iterative process, where the variable is the modulation exponent *m*.

(2) A curve equal to the sum of a simple exponential plus a constant value was generated; thus, by using the average decay constant *τ*_MED_ of the successive point-to-point decays, the integral is the same as that of the original data curve (Equation (11)).
(11)$$V_{S\left(t\right)}=V_{0\sum }\;{\cdot \;e}^{-\left(t/\tau_{\mathrm{MED}}\right)}+V_{R2},\;\;\sum V_{n}\cdot \;\Delta t_{n}=\int V_{S\left(t\right)}\cdot dt.$$

After analyzing the correlation coefficients with 28 initial decay-curve cases, we found that the cut-off potential of Equation (9) is equal to that of Equation (11), i.e., *V*_0EM_ + *V*_0IP_ + *V_R_* = *V*_0Σ_ + *V*_R2_, and that *V*_0*EM*_ + *V*_0*IP*_ = 1.16·*V*_0*M*_. The difference between the cut-off potentials corresponding to the two equations, *V*_0M_ and *V*_0Σ_ + *V*_R2_, were highly correlated (i.e., *R*^2^ = 0.98) with the residual potential. Thus, the minimum deviation relationship providing the residual potential was given by Equation (12):*V*_R_ = (*V*_0Σ_ + *V*_R2_) − 1.16·*V*_0M_.(12)

This decomposition was found to be effective for all the decay curves analyzed in the campaign, except in one case (the first station of sounding P1-N), which had a modulation exponent of 0.55; therefore, the value of 0.6 can be considered as a safety criterion for the modulation exponent, the maximum value obtained for the modulation exponent was 1.0 (corresponding to *V*_R_ = 0).

(3) By subtracting the residual potential *V*_R_, a *V*_IPEM_ (*t*) curve is obtained, in which the separation between the two remaining exponentials is subtler. For this separation, after obtaining an initial potential *V*_01_ by fitting to a modulated exponential $V_{\mathrm{IPEM}}\left(t\right)=V_{01}\cdot e^{-{\left(t/\tau_{1}\right)}^{n}}$, and a decay period τ_2_ by approximating to a simple exponential $V_{\mathrm{IPEM}}\left(t\right)=V_{02}\cdot e^{-\left(t/\tau_{2}\right)}$, the cut-off potential and the polarization period were obtained by approximating the expression Equation (13):(13)$$V_{\mathrm{IPEM}}\left(t\right)=\left(V_{01}-V_{0\mathrm{IP}}\right){\;\cdot \;e}^{-\frac{t}{\tau_{\mathrm{EM}}}}+V_{0\mathrm{IP}}{\;\cdot \;e}^{-\frac{t}{\tau_{\mathrm{IP}}}}.$$

The following values were taken as the initial iteration values: V*_0EM_* = V*_0IP_* = ½V_01_, *τ*_IP_ = *τ*_2_, and *τ*_EM_ = 0.33·τ_2_. By considering that *V*_IPEM_(0) = *V*_0EM_ + *V*_0IP_ = *V*_01_, the values of *V*_0IP_ and *τ*_IP_ were obtained, and thus the final polarisation curve (Equation (14)):(14)$$V_{\mathrm{IP}}\left(t\right)=V_{0\mathrm{IP}}\cdot {\;e}^{-\frac{t}{\tau_{\mathrm{IP}}}}.\;$$

In this iterative process, the relative errors were very low (i.e., between 0.1 and 1.5%), with a difference related more to the small curvature changes than to the difference between the model and the *V*_IPEM_(*t*) values; therefore, the similarity in the second derivative was adopted as the iteration criterion. Figure 3 shows an example of the two curves of *V*_01_ and *V*_02_ obtained with very low error for the data of a decay curve, which nevertheless show a clear difference in curvature, such that the ratio between the two should be centred around 1.

Using the described procedure, for each AB position in each sounding, the decay curves were modelled to obtain the values of *V*_0IP_ and *τ*_IP_. Figure 4a,b show two examples of the decomposition of the decay curves at one of the case-study points. In the first example, the electric field is mostly above the freshwater/saltwater interface, showing a clear positive coupling; in the second example, where a larger fraction of the electric field penetrates below the marine interface, *V*_0EM_ and *V*_0IP_ decrease, showing an apparent induction opposite to that of the created field.

After analysis of the results, we found that the mean value of τ above and below the interface ranged from 1.4 to 1.2 s; based on this small difference, we dismissed the use of decay time in this study.

### 3.3. VES-IPS Inversion

The apparent-resistivity data of each VES-IPS were interpreted by the semi-automatic fitting of a curve generated from an initial model using the software developed by our research group. The method used for generating this curve is based on the convolution of the thicknesses and resistivities using the filter of Ghosh [63].

Furthermore, the apparent-polarisability data were interpreted using the same methodology by varying the mathematical process that generated the curves. For this purpose, we adopted the expression proposed by Seigel [7] and modified by Roy and Poddar [64] for *n* layers:(15)$$P_{a}\left(r\right)=\frac{\rho_{a}\left(r\right)\left(\rho_{j}+P_{j}\;\cdot \;\rho_{j}\right)-\rho_{a}\left(r\right)\left(\rho_{j}\right)}{\rho_{a}\left(r\right)\;{\left(\rho_{j}\right)}^{}};\;\left(j=1,\ldots ,n\right),$$
where: *ρ_a_*(*r*) is the apparent resistivity for the half-space of the *r* electrodes, *ρ*_j_ is the resistivity of layer j, *P*_j_ is the polarisability of layer j.

The program developed starts from the digitized data at progressive distances from 1.5 to 220.0 m, so six points are used in each logarithmic cycle.

## 4. Application for Investigating the Marine Intrusion in the Almuñécar Coastal Aquifer 

### 4.1. Overview of the Study Area and Field Survey

The investigated aquifer is close to the mouth of the Río Verde (Almuñécar) in the province of Granada, located at the SE Iberian Peninsula and limited to the south by the Mediterranean Sea (Figure 5); it comprises loose alluvial conglomeratic and sandy materials, with more abundant silty clay intercalations towards the coast. Below the siliciclastic basin is an impermeable substratum of schist and/or carbonate type. The overexploitation of this area is known from historical hydrochemical data [65]. During the summer period, this overexploitation of water generates an inversion of the subterranean hydraulic gradient, giving rise to a marine intrusion, which for the same reason, allows desalination during the recharge period [66]. The alteration of these processes has been observed for decades. The water quality of the Almuñécar aquifer is magnesium-calcium bicarbonate [67]. The aquifer is characterized by a high hydraulic diffusivity, which is why both the advance and retreat of the saline front are rapid, although the recovery periods have become shorter over the years [68].

Four vertical electrical parametric soundings and 18 vertical electrical and IP soundings (VES-IPS) were performed; their locations are shown in Figure 5. In all cases, a Schlumberger array was used, with interelectrode distances from 1.5 m to a final distance (AB) of 440.0 m, to obtain 10 points per logarithmic cycle. The fieldwork began with a first campaign of four parametric soundings (at each point, two soundings were conducted by placing the electrodes along two perpendicular directions) to evaluate the influence of the layer dip. This means that it was unknown whether the 1D inversion model was applicable or, conversely, the dip of the strata produced very different inversions. Results indicated that the 1D inversion was sufficiently valid. Subsequently, we conducted two temporally separated campaigns with nine VES-IPS, one in October (mid- autumn) and one in February (late winter), to assess the seasonal advance of the marine intrusion. Between October and February is when most of the rainfall occurs in the Almuñécar area. At the beginning of autumn, the driest period has just ended, and at the end of February, the wettest period is coming to an end. Mid-autumn is the date when the marine interface is considered not to have been displaced by the arrival of freshwater, and late winter is the date when the freshwater front is closest to the coast.

Despite the combination of structures, particularly the abundance of greenhouses, it was possible to conduct a line of soundings that provided a reliable delineation of the marine-intrusion interface. In addition, by accessing some piezometers installed in the area, we were able to cross-check the results of the electrical soundings.

In the Almuñécar aquifer, a network of conductivity/temperature sensors was installed on 14 piezometers that were distributed along the axis of the aquifer (Table 1); the instrumentation was used to contrast the degree of advance of the saline front in each campaign (mid-autumn and late winter) [69]. The piezometric network was initially installed by the Spanish Geological and Mining Survey (IGME), which was subsequently managed and extended in successive campaigns by the Confederación Hidrográfica del Sur de España (CHSE) [70].

### 4.2. Analysis of the VES-IPS Results

Figure 6, Figure 7, Figure 8, Figure 9, Figure 10 and Figure 11 present some of the results of the VES-IPS curves obtained during the two campaigns, together with the models obtained after inversion. Given the influence on the final inversion result of the previous model required, the following me issues that have been considered to establish in these previous models are described.

(1) The presence of different resistivity levels was considered when establishing the polarisability levels, despite the SIP curves not showing equally significant differences. We note that the initial polarisability value assigned to the clay levels was obtained from the results of the A5-O sounding (Figure 7a), given that this was the one where the clays were closer to the surface and, therefore, allowed a more precise determination of their polarizability (~20 mV/V).

(2) Following Equation (15) [7], the resistivity gradient significantly affected the polarisability values, with the gradients of the IPS depending not only on the difference in P values between layers but also on the difference in resistivity. One of the effects of this interrelationship was the possibility of sharp changes in the IP curve, such as the one for the A5-O sounding and from a sounding conducted in another basin on the same coast. In principle, it could be assumed that there was some error in the data collection (despite repeatedly verifying these results), but the inversion showed that the theoretical model corroborates such an abrupt change. In this regard, we note that similar cases in unpublished marine intrusion work have been found (see Figure 7a,b).

(3) Although the convenience of a greater final AB distance became apparent from the field measurements and more evident after the inversion of the soundings, such an increase could not be achieved because of the difficulty of progressively crossing more crop farms. In any case, the resistivity of the schists was well established in VESs further in the north. However, because of the distribution of polarisability values with depth, the polarisability value was well established from both A4-O VES-IPS and A4-F VES-IPS (Figure 8), especially during the marine-intrusion regression campaign conducted in October (mid-autumn).

Figure 9 shows VES-IPS results with clear differentiation of the geoelectric levels; however, the values of the last level are somewhat undefined (in A8-O VES-IPS and A9-O VES-IPS, the polarisability is well-defined). Fortunately, the inversion provided values more accurately, given its influence on the behaviour of the final part of the VES-IPS curves.

(4) For the VES-IPS inversion, it was necessary to use models with an intermediate resistivity layer between the layer saturated with freshwater and the layer saturated with saltwater. In A3-O VES-IPS (Figure 10), the resistivity of this layer was well established, with a value of 9 Ω·m; however, this does not imply that this value should be maintained throughout the section obtained in each survey. In some cases, a progressively decreasing resistivity model could have been used, but this would unnecessarily complicate the VES modelling.

(5) The lack of accuracy in deeper levels (i.e., before reaching the schist substrate) with different resistivities and reduced thicknesses must be considered. In some cases, the subdivision into distinct levels is strongly supported by the differentiation shown by the IPS curves. Figure 9 shows the A9-O VES-IPS curves, in which the possibility of the previous model predicting the value of the geoelectric level corresponding to the gravel/conglomerate interface with that interstitial fluid has been reduced.

(6) The inversion would produce important differences in the obtained distribution of thicknesses and/or resistivities if the initial models did not consider the presence of the clay layer. As an example, Figure 11 shows the inversion of the A5-F sounding without consideration of the clay layer, in which the greatest difference occurs in the depth of the substratum, which is 96 m, as opposed to 65 m obtained in the inversion of the same sounding with consideration of the clay layer (Figure 6).

By comparing the results of the lithological columns in the two boreholes located very close to the delineated profile with those of the geoelectric soundings, we observed that the latter showed a somewhat more differentiated subdivision of the subsoil, determining up to six differentiated geoelectric levels. Table 2 shows the final inversion results obtained in the VES-IPS set mid-autumn.

### 4.3. Seasonal Variation of the Freshwater/Saltwater Interface

Based on the VES-IPS inversion results in the two campaigns conducted, we established the resistivity ranges used for lithological assignment (Table 3), thereby allowing the determination of the seasonal variation of the freshwater/saltwater interface.

Although the average resistivity value for gravels/conglomerates that were saturated with freshwater was 230 Ω·m, this value was not considered as an initial model because it varied strongly in the area. The resistivity range of the schist substratum was also maintained. The most critical aspects of this model assignment were the low thickness levels located at increased depths, making it difficult to discriminate between the interface and intrusion zones. Regarding the polarisability values, the gravel layers showed a mean polarisability of 6 mV/V regardless of the characteristics of the saturation fluid; however, while the clay levels showed a higher mean polarisability value of 20 mV/V, the schistose substrate had a markedly higher mean polarisability value of 75 mV/V.

Using these ranges, the freshwater–saltwater interface was mapped for two seasons, i.e., mid-autumn and late winter. Figure 12 shows the sections obtained from the VES-IPS, revealing the marine intrusion regression in late winter and a somewhat smaller regression of the interface zone. The red dashed lines indicate the coastal line and the area where the resistivity distribution has been extrapolated to have a spatial representation under the sea.

## 5. Discussion

As already mentioned, the Almuñécar aquifer was chosen because the problem of intrusion has been studied since the early 1980s [71,72,73,74,75], which allows us to contrast and reinforce the results obtained. From the measurements in the piezometer network, it is known that the maximum values of conductivity measured in the piezometer network occur between the months of October and November [76]; this supports the fact that the intrusion is maximum in these months. The characterization of the area cannot rely exclusively on the information obtained from the boreholes because of its high heterogeneity due to the frequent intercalations of fine-grained materials [77]. Plata and Rubio [16] established a new geological characterization combined with information from boreholes and the application of geophysical techniques (magnetotelluric and vertical electrical soundings), which is coherent with the results obtained in this study. These authors mention that problems were encountered in the discrimination of some levels, finding inconsistencies with the lithological columns of the boreholes. The combination of resistivity and polarisability measurements, and the application of the methodology proposed in this study, allow these difficulties to be reduced.

Regarding the methodology developed, the most important question to be addressed in this analysis is, when is it necessary to perform decay curve decomposition? In other words, when should the EM and residual potential effects be removed?

Cases involving EM effects with small decay times (of the order of the smallest value of the measurement range) are the ones that initially seem to require such decomposition the most. Interestingly, the cases in which the polarisation effect is low (~2 mV/V) are clearer.

To determine when a residual potential value needs to be eliminated, it is sufficient to obtain a value that minimizes the approximation (by least squares) of a simple exponential to the data curve. If this value is negligible (i.e., <5%) compared to the cut-off value at the origin, it is not necessary to consider it.

After the first—and not definitive—removal of the residual potential, the resulting curve appears in most cases to be a simple exponential. This is because when the ratios between the parameters of the two resulting curves are less than 5, the sum of exponentials presents a behaviour very similar to that of a simple exponential; in fact, its approximation to an exponential reaches a regression coefficient of up to 0.9999. After analyzing the cases studied, we found that decomposition should be conducted whenever the curve of the logarithms of the measured values is closer to an exponential than to a straight line, i.e., whenever the ratio between the regression coefficients of both is less than 1.

Note that the discrimination conducted has been effective, given that the study depth was not increased (i.e., approximately 100 m), with the EM effects in the study window (i.e., 0.5–3.5 s) being lower or similar to those of polarization. A similar result was observed in relation to the residual potential, which was below 50% of the potential value at *t* = 0^+^.

Regarding the obtained residual potential *V_R_*, we note that although it has been attributed to the possible polarisation of the electrodes, the comparative analysis of the obtained values showed the existence of a strong correlation (correlation coefficient: 0.96) between this potential and the polarisation of the medium. Even non-polarisable potential electrodes deviate from their expected behaviour by an amount proportional to the polarisation potential exerted on them by the subsurface. However, we note that there may be a relationship between *V_R_* and ground behaviour.

Regarding the electromagnetic-effect cut-off potential *V*_0*EM*_ (and its unique time constant), it is important to note that it is futile searching for a direct interpretation of the resulting values, as this has been obtained by analyzing the decay curve, whose effect is already very small and its evolution towards zero has no particular significance. Moreover, assigning only the first section of the decay curve to an induction effect in the subsurface would be problematic, as this value may include other effects, such as coupling between the cables.

Finally, note that the developed methodology should be of interest to other researchers since it is easily applicable, especially considering that the number of measurement points used for analyzing the decay curve is the minimum that can be obtained with contemporary equipment.

### Error Analysis

One of the main reasons for developing this work is the presence of uncertainties due to noise and errors in the measurements. These uncertainties do not allow further subdivision of the decay curve as it is done in the Laplace decomposition of the IP curves.

First, consider the error resulting from the correct layout of the electrodes in the linear arrays. In the data of this campaign, this error, relating to the measurements at the junction points due to the increase of the MN distance (using between 2 and 4 points), was, on average, 15% of the resistivity value. We note that this value did not affect the distribution of the layers that we were able to obtain through the soundings.

This error likely did not affect the values of the IP decay curves at each station, as the measurement procedure was performed by adopting valid values when the stacking process did not produce significant changes in the result. However, the polarisability values obtained from the control measurements indicated that the measured values involved uncertainties. Although the average error resulting from the values of the decay curve versus time was small (i.e., 0.05 mV/V), its relative value was not, with an average value of 4.0%. This uncertainty requires considering that selecting the segments between each pair of points on the curve may generate different decay processes corresponding only to measurement errors.

Another error requiring analysis is the one owing to the process used for curve decomposition. In this regard, the comparison between the results of the lithological section of the study areas and the position of the saline interface (by means of the columns and the water analysis of the piezometers in the study area) revealed coherence.

However, the verification of the numerical value of this error cannot be performed simply through this test, because the different values of polarisability and deduced time constants would be equally coherent within a clearly higher assignment range. In other words, errors of up to 15% in the values of these parameters would not affect the lithological and hydrochemical assignments. Therefore, to estimate this error, the values obtained from repeated control measurements were compared again, and it was an average relative error of 3.9% for polarisability and 4.1% for the decay constant.

## 6. Conclusions

In this study, we proposed a method for processing the induced polarisation decay curve of the subsurface to obtain polarisability. Following the generic analysis of the PI decay curves shown in the introduction section (see Figure 1), it can be concluded that our method is more appropriate and decisive than using the conventionally obtained chargeability since the ‘average’ chargeability results in loss of information regarding how different strata depolarise and the effect that certain lithologies have on EM coupling.

Compared to the published geophysical surveys in which the combination with IP was not used, the results obtained in this study revealed the advantages of using joint geophysical techniques to avoid indeterminacy regarding the interpretation while verifying that knowledge of polarisability constitutes an important contribution regarding the resistivity results. This combination may be of interest regarding the geological assignment of physical parameters in general, but it is even more necessary in investigations of the salinization of coastal aquifers, both with regard to basic monitoring but also prospecting for uncontaminated levels. The inclusion of the clay layers, established mainly from their polarizability values, provides models that improve the discrimination of the state of marine intrusion. The inclusion of polarisability significantly modifies the resistivity distribution obtained with the SEVs, allowing the detection of resistivity variations due to the change in formation fluid conductivity.

Intrinsically related to the previous conclusion, it can also be stated that the inclusion of a transition zone between the purely saline front and the continental freshwater water facilitates the hydrochemical interpretation obtained with the electrical survey.

The very extensive network of piezometers in the aquifer studied allows us to affirm that the methodology applied has provided a sufficiently precise determination of this advance, and its application in other areas would allow the number of piezometers to be reduced and the location of each borehole to be optimized.

The obtained seasonal variation of the frontal position of the studied marine intrusion confirms the possibilities of the referred combination, even if the thicknesses of the strata are close to the resolution capacity of the 1D techniques.

The proposed methodology is applicable in other time domain-induced polarisation studies applied to other different problems.

## Figures and Tables

**Figure 1 sensors-23-00708-f001:**
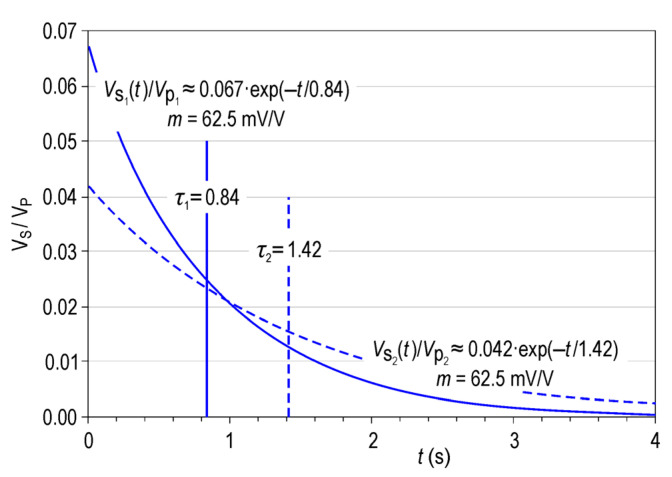
Example of two decay curves with the same chargeability but different polarisabilities and decay times.

**Figure 2 sensors-23-00708-f002:**
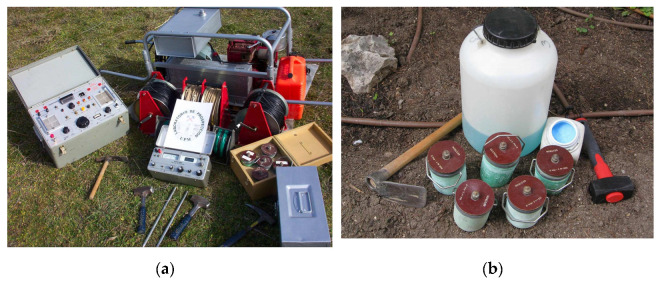
(**a**) Scintrex TSQ-3 equipment and IPR-10A receiver. (**b**) Fat non-polarisable electrodes with CuSO_4_.

**Figure 3 sensors-23-00708-f003:**
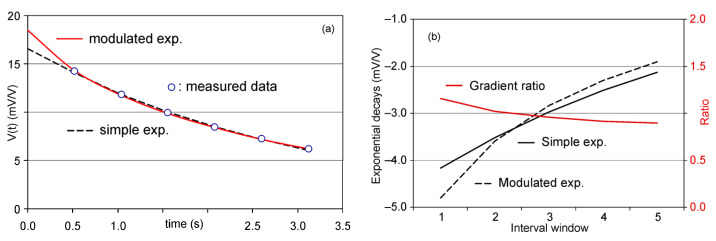
(**a**) Approximation exponentials. (**b**) Derivative curves and their ratios.

**Figure 4 sensors-23-00708-f004:**
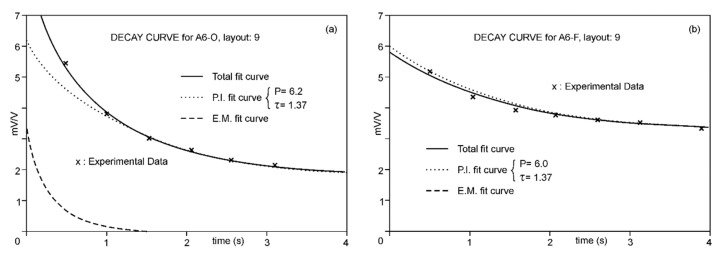
(**a**). Decay curve of sounding A6-O, station 9. (**b**). Decay curve of sounding A6-F, station 9.

**Figure 5 sensors-23-00708-f005:**
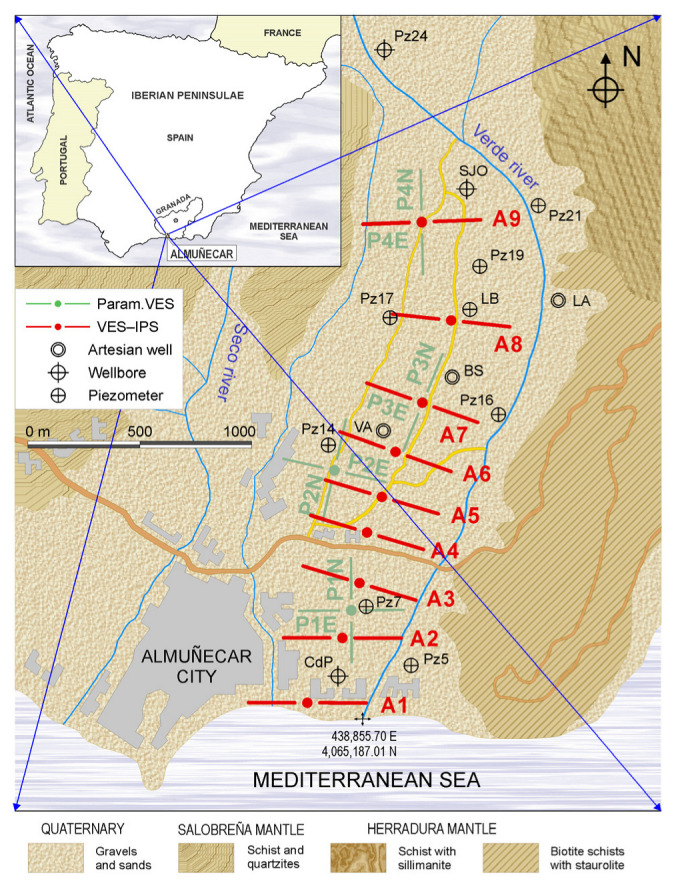
Location of the study area, a sketch of the locations of the soundings, and geology of the area. The coordinates of the river’s mouths are Seco River 437,720.62 m E & 4,065,188.01 m N, Verde River 438,855.70 m E & 4,065,187.01 m N.

**Figure 6 sensors-23-00708-f006:**
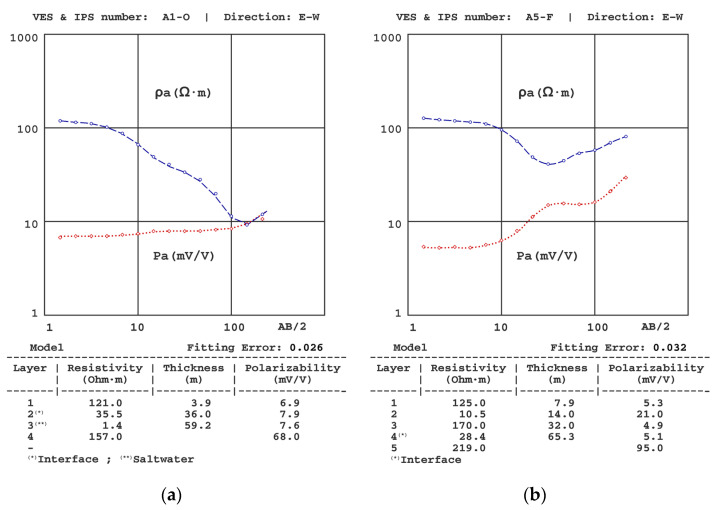
Results of the (**a**) A1-O VES-IPS obtained mid-autumn and (**b**) A5-F VES-IPS obtained late winter.

**Figure 7 sensors-23-00708-f007:**
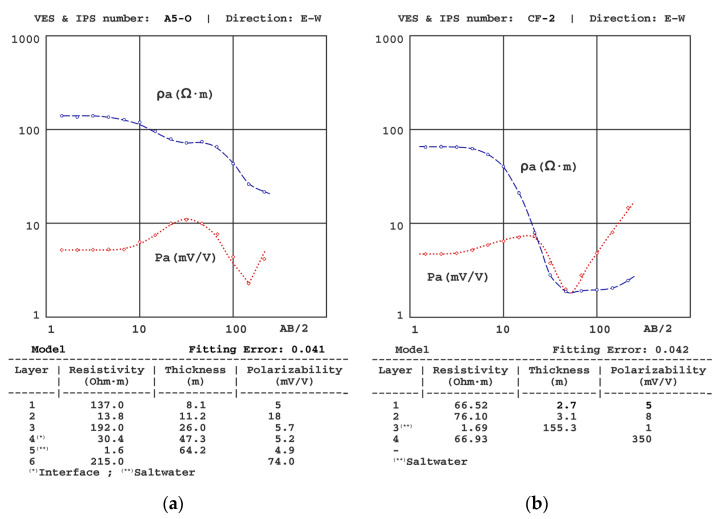
Results of the (**a**) A5-O VES-IPS obtained mid-autumn and (**b**) of the CF2 VES-IPS conducted in another basin in the same region.

**Figure 8 sensors-23-00708-f008:**
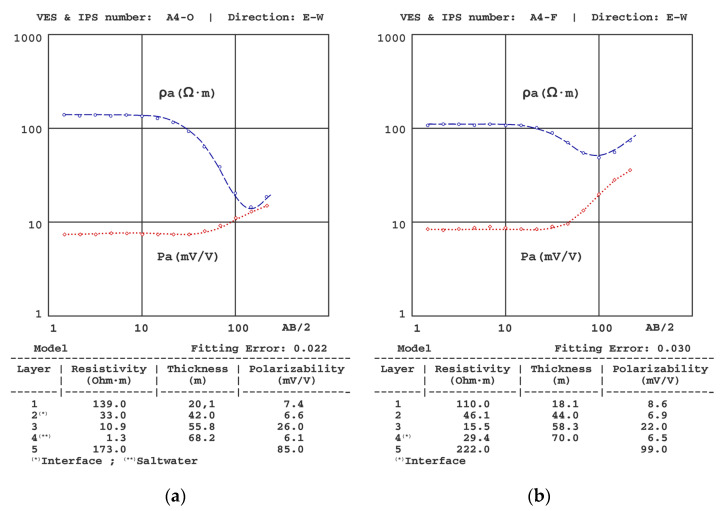
Results of the (**a**) A4-O VES-IPS obtained before winter and (**b**) A4-F VES-IPS obtained after winter.

**Figure 9 sensors-23-00708-f009:**
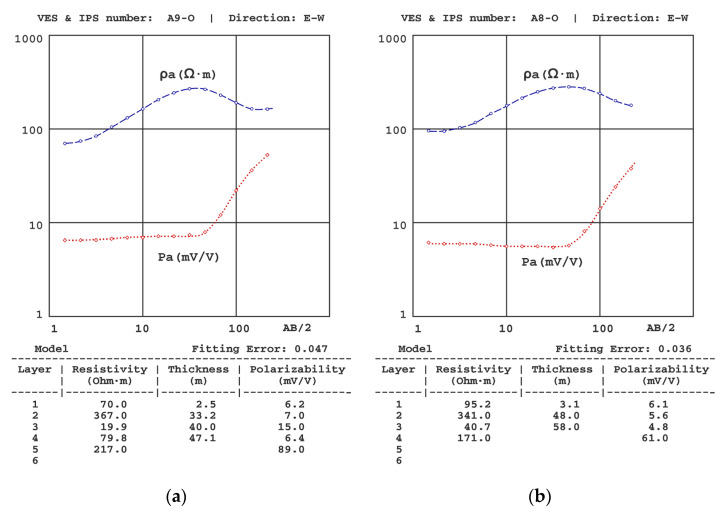
Results of the (**a**) A9-O VES-IPS and (**b**) A8-O VES-IPS were both obtained mid-autumn, showing a clear differentiation of the geoelectric levels and an apparent uncertainty of the values of the deeper layers.

**Figure 10 sensors-23-00708-f010:**
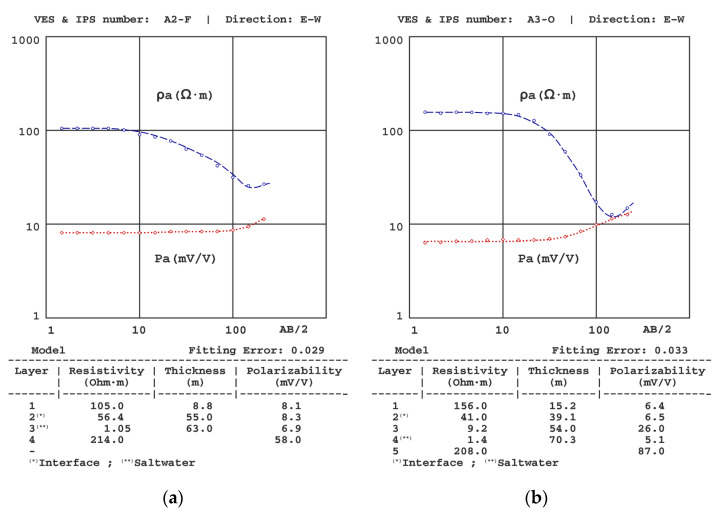
Results of the (**a**) A2-F VES-IPS and (**b**) A3-O VES-IPS, both obtained mid-autumn.

**Figure 11 sensors-23-00708-f011:**
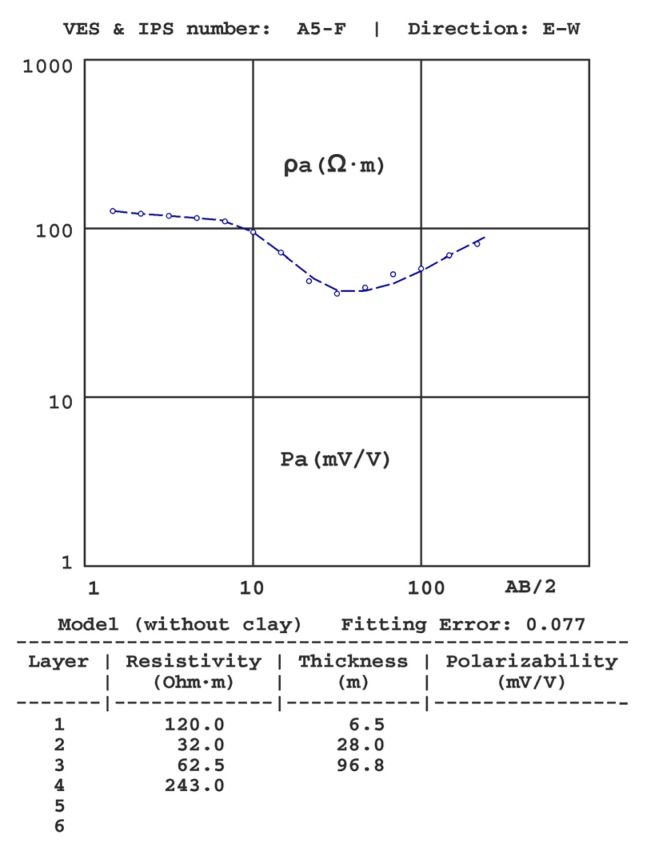
Results of the A5-F VES-IPS were obtained in late winter.

**Figure 12 sensors-23-00708-f012:**
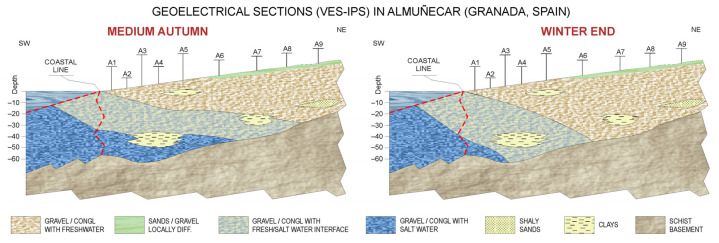
Sections showing the marine intrusion progress in two different seasons.

**Table 1 sensors-23-00708-t001:** Temperature and conductivity values were logged in the month of October.

Sondeo	T (°C)	σ (μS/cm)
CDP	16.9	31,504
Pz5	15.4	18,701
Pz7	17.4	4063
Pz14	17.4	23,551
VA	18.8	2868
Pz16	18.7	6767
BS	19.2	13,546
Pz17	19.6	2056
LB	21.9	6686
LA	25.5	1249
SJO	22.3	1796
Pz21	19.07	934
Pz24	18.4	882
Pz19	20.2	5546
CDP	16.9	31,504

**Table 2 sensors-23-00708-t002:** Resistivity and polarisability values in October (mid-autumn) were obtained with VES-IPS inversion.

A1			A2			A3			A4			A5		
z	*ρ*	*P*	z	*ρ*	*P*	z	*ρ*	*P*	z	*ρ*	*P*	z	*ρ*	*P*
3.9	121	6.9	9.5	132	6.4	15	156	6.4	20	139	7.4	8.1	137	5.1
36	35.5	7.9	41	47.9	7.0	39	41	6.5	42	33.0	6.6	11	13.8	18
59	1.36	7.6	60	0.95	6.5	54	9.0	26	56	10.9	26	26	192	5.7
--	157	68	--	174	87	70	1.4	5.1	68	1.3	6.1	47	30.0	5.2
					--	208	87	--	173	85	64	1.6	4.9	
												215	74	
A6			A7			A8			A9					
z	*ρ*	*P*	z	*ρ*	*P*	z	*ρ*	*P*	z	*ρ*	*P*			
2.7	112	6.7	3.8	155	7.2	3.1	95.2	6.1	2.5	70.3	6.2			
32	241	5.7	38	312	7.9	48	341	5.6	33	367	7.0			
49	37	6.3	45	10.2	20	58	40.7	4.8	40	19.9	15			
60	1.5	5.5	59	41.5	6.3	--	172	61	47	79.8	6.4			
--	171	91	--	193	87	--				217	89			

**Table 3 sensors-23-00708-t003:** Resistivity and polarisability ranges used for lithological assignment and mean values when applicable.

*ρ*→	121	367	Gravels/conglomerates Fresh water	30	48	Gravels/conglomerates Interface	1.0	1.6	Gravels/conglomerates Salt water	9.0	14	Clay	157	217	Schist (Substrate)
				40			1.2			10					
P→	5	10		5	10		5	10		15	25		60	90	
	6			6			6			22			75		

## Data Availability

The data presented in this study are available on request from the corresponding author. The data are not publicly available due to some special reasons.

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
