# Peer review of "A Different Processing of Time-Domain Induced Polarisation: Application for Investigating the Marine Intrusion in a Coastal Aquifer in the SE Iberian Peninsula"

_sensors, 2023, doi:10.3390/s23020708_

Round 1
Reviewer 1 Report
Many thanks for giving me the opportunity to review the manuscript of Jesús Díaz-Curiel et al. titled Different processing of time-domain-induced polarisation: Application for investigating the marine intrusion in the SE Iberian Peninsula. This study proposed a method for processing the induced polarisation decay curve of the subsurface to obtain polarisability. The proposed method is appropriate and decisive. Overall, this is a good paper, in addition to the fact that it needs to be reorganized, especially the discussion and conclusion parts. My comments are listed here:
1. The authors need to rewrite the discussion part. The discussion form in this part is unappreciated. It is incredible that you have not cited any reference in the text. You need to compare your results with the previous studies and discusses the advantage of your method as well as the core finding of your results regarding marine intrusion in the SE Iberian Peninsula.
2. There are too many colloquial sentences in the text. Such as "We note that the discrimination that we conducted"; "First, we consider the error resulting from the correct layout"; "Finally, we note that the developed methodology should be of interest to"
3. It seems to me that it is a research paper. So I suggest the authors pay more attention to the results related to the investigation of the marine intrusion in the SE Iberian Peninsula. I cannot see any sentence regarding this part in the conclusion part in the current version. Please focus on your core findings in this study and revise the discussion and conclusions parts.
Author Response
Madrid, December 31st, 2022
Dear Editors and Reviewers,
Following your indications, we attach a detailed response indicating the changes we have made considering all suggestions from the editor and the referees. All of them are marked up using the “Track Changes” function in the new version of the manuscript. We have also replaced the previous figures with new corrected ones. We are confident that we have given a satisfactory response to all the reviewers' suggestions, and we are grateful for the reviews carried out which has allowed us to correct some mistakes and improve some parts.
We would like to clarify that the aim of this work is to propose a different methodology for time domain induced polarisation processing. It is applicable to any polarisation study. As an application case we have chosen a marine intrusion problem, because it seems to us especially useful in environments with low resistivity lithologies. The choice of the Almuñécar aquifer is due to the fact that the problem of intrusion is well known. However, it is not the aim of this work to conduct a generalised study of the intrusion problem in this aquifer, nor to establish the most appropriate parameters for water management in the area.
We have modified the title to make it clearer: “A different processing of time-domain induced polarisation: Application to the investigation of marine intrusion in a coastal aquifer in the SE Iberian Peninsula”
Reviewer 1
Comments and Suggestions for Authors:
1) The authors need to rewrite the discussion part. The discussion form in this part is unappreciated. It is incredible that you have not cited any reference in the text. You need to compare your results with the previous studies and discusses the advantage of your method as well as the core finding of your results regarding marine intrusion in the SE Iberian Peninsula.
Following the reviewer's recommendation, the discussion and conclusions have been modified.
2) There are too many colloquial sentences in the text. Such as "We note that the discrimination that we conducted"; "First, we consider the error resulting from the correct layout"; "Finally, we note that the developed methodology should be of interest to"
Thank you for this suggestion. Considering this reviewer comment, we have substituted the majority of “we” appearing in the manuscript in order to make it less colloquial.
3) It seems to me that it is a research paper. So I suggest the authors pay more attention to the results related to the investigation of the marine intrusion in the SE Iberian Peninsula. I cannot see any sentence regarding this part in the conclusion part in the current version. Please focus on your core findings in this study and revise the discussion and conclusions parts
Thank you for this suggestion. Considering this reviewer comment, we have added some references in the discussion, and in the references list.
Reviewer 2 Report
Dear Sir/Madam,
I just went through the submitted document, and although I am not an expert in this field, I found it very and inovating. It provides a new methodology on the interusion of salt water in aquifers, which I believe that it would be in the interest of the international scientific community.
Best regards,
Author Response
Madrid, December 31st, 2022
Dear Editors and Reviewers,
Following your indications, we attach a detailed response indicating the changes we have made considering all suggestions from the editor and the referees. All of them are marked up using the “Track Changes” function in the new version of the manuscript. We have also replaced the previous figures with new corrected ones. We are confident that we have given a satisfactory response to all the reviewers' suggestions, and we are grateful for the reviews carried out which has allowed us to correct some mistakes and improve some parts.
We would like to clarify that the aim of this work is to propose a different methodology for time domain induced polarisation processing. It is applicable to any polarisation study. As an application case we have chosen a marine intrusion problem, because it seems to us especially useful in environments with low resistivity lithologies. The choice of the Almuñécar aquifer is due to the fact that the problem of intrusion is well known. However, it is not the aim of this work to conduct a generalised study of the intrusion problem in this aquifer, nor to establish the most appropriate parameters for water management in the area.
We have modified the title to make it clearer: “A different processing of time-domain induced polarisation: Application to the investigation of marine intrusion in a coastal aquifer in the SE Iberian Peninsula”
Reviewer: 2
Comments and Suggestions for Authors:
I just went through the submitted document, and although I am not an expert in this field, I found it very and innovating. It provides a new methodology on the intrusion of salt water in aquifers, which I believe that it would be in the interest of the international scientific community.
We want to thank this reviewer for the kindly words about our work.
Reviewer 3 Report
Revision of the manuscript: Different processing of time-domain-induced polarisation: Application for investigating the marine intrusion in the SE Iberian Peninsula
The manuscript presents the developments regarding the time-domain-induced polarisation method, as a supporting tool for resistivity soundings during investigations of coastal detrital aquifers that are salinized by the marine intrusion.
The text is well written and the topic could be interesting to the readers of the journal: Sensors.
The method is well explained but there is a shortcoming in the interpretation of the presented example. Not enough information is given on the study area. For example, there are several wells in the area and nearby, but no information is used to discuss your findings. What is the water quality in that wells? In which year did you obtain the data? The map of the study area has no scale, the legend is incomplete. Figures 7-11 should include the position of the interface and the position of the saltwater in each of the sections. Finally, from table 2 it can be concluded that the interpretation given is based only on differences in resistivity, but not on polarization.
The whole of chapter 4. Application for investigating the marine intrusion in the SE Iberian Peninsula
should be strengthened to publish the article.
Minor observations are:
Line 23-24
You write: …, one before and one after winter (i.e., in October and February, respectively).
February is still Winter so it would be better to write only the month.
Please explain here why wintertime is so important for changes in the seawater intrusion in the study area.
Line 25 26
You write: The results reveal the position of the saline front during each campaign, while reflecting the seasonal movement of the marine intrusion.
Please describe what are the findings achieved by this investigation. What changes in water management were suggested, based on the findings?
Line 146
You write: The IP prospecting method was initially called over-voltage nd became very ….
Should be:
The IP prospecting method was initially called over-voltage and became very ….
Line 203
You write: …using filters for noninfinite signals…
Should be:
…using filters for non-infinite signals…
Figure 5
Line 400 401
This map needs to include coordinates and a scale. It is also unclear which part of Granada corresponds to the study area.
Line 402 -405
In the Almuñécar aquifer, a network of conductivity/temperature sensors was installed on 15 piezometers that were distributed along the axis of the aquifer; the instrumentation was used to contrast the degree of advance of the saline front in each campaign (before and after the winter) [67…
The citation number 67 is incomplete, please give the full internet direction to access the text.
Please give more information obtained from the 15 piezometers.
Please include in the legend: what do the green points and P 1 to P4 indicate.
Please give more information on the artesian well and the wellbore, both located in the study area.
67. Díaz-Curiel, J.; Martín Sánchez, D.; Maldonado Zamora, A.; Gómez Martos, M. Red de control σ/T para el estudo de intrusion marina en Almuñécar (Granada). Boletín Geológico y Min. 1995, 106(4), 358-372. Instituto Geológico y Minero de España (http://www.igme.es/internet/pr...). ISSN 0366-0176.
Should be: 67. Díaz-Curiel, J.; Martín Sánchez, D.; Maldonado Zamora, A.; Gómez Martos, M. Red de control σ/T para el estudio ...
Line 487
Table 2. Resistivity and polarisability ranges used for lithological assignment, and mean values
when applicable.
In your example of the application of the method, please explain why polarisability was useful to define the position of saltwater intrusion as in the table there is no difference of ranges used for lithological assignment, and mean values in Gravels/conglomerates with freshwater, Gravels/conglomerates with Interface, and Gravels/conglomerates with salt water. As it seems your interpretation is based only on differences in resistivity.
Figure 12 line 500
Please indicate what do the red dashed lines indicate.
Please explain what layer the green color signifies.
Line 537, 538
.. particular significance. Moreo-
ver, assigning …
Should be:
.. particular significance. More-
over, assigning …
Line 570, 571
not affect the litholog-
ical and hydrochemical assignments
Line 574.
The Conclusions are written more like an abstract, here you should discuss if the objectives were achieved, and how your findings could be useful in general.
Author Response
Madrid, December 31st, 2022
Dear Editors and Reviewers,
Following your indications, we attach a detailed response indicating the changes we have made considering all suggestions from the editor and the referees. All of them are marked up using the “Track Changes” function in the new version of the manuscript. We have also replaced the previous figures with new corrected ones. We are confident that we have given a satisfactory response to all the reviewers' suggestions, and we are grateful for the reviews carried out which has allowed us to correct some mistakes and improve some parts.
We would like to clarify that the aim of this work is to propose a different methodology for time domain induced polarisation processing. It is applicable to any polarisation study. As an application case we have chosen a marine intrusion problem, because it seems to us especially useful in environments with low resistivity lithologies. The choice of the Almuñécar aquifer is due to the fact that the problem of intrusion is well known. However, it is not the aim of this work to conduct a generalised study of the intrusion problem in this aquifer, nor to establish the most appropriate parameters for water management in the area.
We have modified the title to make it clearer: “A different processing of time-domain induced polarisation: Application to the investigation of marine intrusion in a coastal aquifer in the SE Iberian Peninsula”
Reviewer: 3
Comments and Suggestions for Authors:
The method is well explained but there is a shortcoming in the interpretation of the presented example. Not enough information is given on the study area. For example, there are several wells in the area and nearby, but no information is used to discuss your findings. What is the water quality in that wells? In which year did you obtain the data? The map of the study area has no scale, the legend is incomplete. Figures 7-11 should include the position of the interface and the position of the saltwater in each of the sections. Finally, from table 2 it can be concluded that the interpretation given is based only on differences in resistivity, but not on polarization.
We thank the reviewer for his suggestions. The information on the study area and the information on the wells has been expanded, the figures have been modified, the rest of the comments are answered in more detail below.
1.a) Line 23-24
You write: …, one before and one after winter (i.e., in October and February, respectively).
February is still Winter so it would be better to write only the month.
The reviewer is right, in February it is still winter, the proper way is end of winter. We have corrected this in the text. We have referred to the campaigns in the text as mid-autumn and late winter.
1.b) Please explain here why wintertime is so important for changes in the seawater intrusion in the study area.
Between October and February is when most of the rainfall occurs in the Almuñécar area. At the beginning of autumn, the driest period has just ended, and at the end of February the wettest period is coming to an end. With mid-autumn being the date when the marine interface is considered not to have been displaced by the arrival of freshwater, and late winter being the date when the freshwater front is closest to the coast.
We have added this explanation in the manuscript, in section 4.1.
2) Line 25 26
You write: The results reveal the position of the saline front during each campaign, while reflecting the seasonal movement of the marine intrusion.
Please describe what are the findings achieved by this investigation. What changes in water management were suggested, based on the findings?.
The most important finding is the methodology itself. The use of polarisability instead of chargeability. The knowledge of polarisability is an important contribution to the resistivity results. This combination may be of interest in terms of geological assignment.
Following the reviewer's recommendation, we have modified the discussion and conclusions parts.
Although it is not the purpose of this study to deal with water management, as has already been mentioned, the problem of intrusion in this aquifer is well known. One of the biggest problems is found in the unregulated extraction of water from wells. The study of the intrusion is conducted not only by induced polarisation, but also by a conductivity and temperature control network, anticipating the advance of the intrusion by at least 9 days. This allows to stop extractions and minimise the contamination of the aquifer.
3) Line 146
You write: The IP prospecting method was initially called over-voltage nd became very ….
Should be:
The IP prospecting method was initially called over-voltage and became very …..
The only difference we see between the two sentences is "nd" and "and", but in the manuscript it is spelled correctly. We do not know if the reviewer is referring to another issue.
4) Line 203
You write: …using filters for noninfinite signals…
Should be:
…using filters for non-infinite signals…
Thank you for this comment. We have corrected it in the manuscript.
5) Figure 5
Line 400 401
This map needs to include coordinates and a scale. It is also unclear which part of Granada corresponds to the study area
Following the reviewer's recommendations, we have modified figure 5, adding the coordinates of the mouth of the Seco River and the Verde River in the figure caption. The study area is located in the coast of Granada, more specifically between Almuñécar town and the Verde River, as indicated. The legend has been modified to include the vertical parametric electrical soundings (designated with green colour), the scale is located below the legend.
6.a) Line 402 -405
In the Almuñécar aquifer, a network of conductivity/temperature sensors was installed on 15 piezometers that were distributed along the axis of the aquifer; the instrumentation was used to contrast the degree of advance of the saline front in each campaign (before and after the winter) [67…
The citation number 67 is incomplete, please give the full internet direction to access the text.
Sorry for this mistake in the citation. Corrected reference has been included in the References section.
6.b) Please give more information obtained from the 15 piezometers.
The position of 14 of the piezometers has been included in figure 5, one of them has been omitted as it is already far away from the study area. A table with conductivity values logged in the piezometers has also been included in the manuscript, in section 4.1 .
More information about the piezometers has been included in section 4.1.
6.c) Please include in the legend: what do the green points and P 1 to P4 indicate.
The green points named P1 to P4 are the parametric vertical electrical soundings, the legend has been modified to include them (response 5).
6.d) Please give more information on the artesian well and the wellbore, both located in the study area.
Already answered in response 6.b.
6.e) 67. Díaz-Curiel, J.; Martín Sánchez, D.; Maldonado Zamora, A.; Gómez Martos, M. Red de control σ/T para el estudo de intrusion marina en Almuñécar (Granada).Boletín Geológico y Min.1995, 106(4), 358-372. Instituto Geológico y Minero de España (http://www.igme.es/internet/pr...). ISSN 0366-0176.
Should be: 67. Díaz-Curiel, J.; Martín Sánchez, D.; Maldonado Zamora, A.; Gómez Martos, M. Red de control σ/T para el estudio ....
We thank the reviewer for his correction. We have modified it (response 6.a).
7) Line 487
Table 2. Resistivity and polarisability ranges used for lithological assignment, and mean values when applicable.
In your example of the application of the method, please explain why polarisability was useful to define the position of saltwater intrusion as in the table there is no difference of ranges used for lithological assignment, and mean values in Gravels/conglomerates with freshwater, Gravels/conglomerates with Interface, and Gravels/conglomerates with salt water. As it seems your interpretation is based only on differences in resistivity
In this study, the inclusion of polarisability is not done with the purpose of differentiating the marine intrusion but because it significantly modifies the resistivity distribution obtained with the SEVs, allowing the detection of resistivity variations due to the change in the conductivity of the formation fluid. In addition, the polarisation values make it possible to discriminate the clay levels from the levels affected by the intrusion. The resistivity of the clay can give rise to confusion and can be interpreted as areas affected by saline water, in this sense it is the polarisation value that helps to differentiate one level from another.
Following the reviewer's suggestion, this issue has been clarified in the modifications made in the discussion and conclusions sections.
8.a) Figure 12 line 500
Please indicate what do the red dashed lines indicate
The red dashed lines represent the coastal line and the area where the resistivity distribution has been extrapolated to have a spatial representation under the sea. Following the reviewer's recommendation, we have included this clarification in section 4.3.
8.b) Please explain what layer the green color signifies.
The layer designated with green colour reflects a shallow layer more affected by local variations of seasonal and/or agricultural origin.
We thank the reviewer for his comment. We have completed the legend with this missing layer in Figure 12.
9) Line 537, 538
.. particular significance. Moreo-
ver, assigning …
Should be:
.. particular significance. More-
over, assigning …
The hyphenation is automated in the MSword ©. To correct this, we have removed the hyphenation.
10) Line 570, 571
not affect the litholog-
ical and hydrochemical assignments
Same as previous response.
11) Line 203
The Conclusions are written more like an abstract, here you should discuss if the objectives were achieved, and how your findings could be useful in general.
Following the reviewer's recommendation, we have modified the conclusions.
Round 2
Reviewer 1 Report
I am pleased to see the major changes the authors have made, and the academic level of this paper has been significantly improved. But the conclusions part is too long. This part needs to be rewritten, and only the main conclusions of the study should be presented in a short section. I suggest that it can be accepted for publication after such a minor revision.
Author Response
Dear Editors and Reviewers,
Following your indications, we attach a detailed response indicating the changes we have made considering all suggestions from the editor and the referees. All of them are marked up using the “Track Changes” function in the new version of the manuscript. We are confident that we have given a satisfactory response to all the reviewers' suggestions.
Reviewer 1
I am pleased to see the major changes the authors have made, and the academic level of this paper has been significantly improved. But the conclusions part is too long. This part needs to be rewritten, and only the main conclusions of the study should be presented in a short section. I suggest that it can be accepted for publication after such a minor revision.
Many thanks to reviewer for his comments on our work. The Conclusions section was modified and expanded in accordance with the indications of reviewer 3 in Round 1. This section is now 390 words long and we do not consider it to be excessively long. We have tried to discuss the achievement of the objectives of our study as well as the future usefulness of our research for other researchers. We have modified part of the conclusions, although without shortening it too much so as not to contravene what was suggested in the previous round.
Reviewer 3 Report
All requested changes were made, only some style check is required before publishing.
Author Response
Dear Editors and Reviewers,
Following your indications, we attach a detailed response indicating the changes we have made considering all suggestions from the editor and the referees. All of them are marked up using the “Track Changes” function in the new version of the manuscript. We are confident that we have given a satisfactory response to all the reviewers' suggestions.
Reviewer: 3
The authors need to rewrite the discussion part. The discussion form in this part is unappreciated. It is All requested changes were made, only some style check is required before publishing.
Thank you to the Reviewer for the comment. According to your suggestion, we have reviewed the entire manuscript and have modified text that was in a different colour, text that was in a different font, and some words and expressions.